# Efficient deep learning model for classifying lung cancer images using normalized stain agnostic feature method and FastAI-2

Pranshu Saxena[1], Sanjay Kumar Singh[2], Mamoon Rashid[3], Sultan S. Alshamrani[4] and Mrim M. Alnfiai[4]

[1] School of Computer Science Engineering and Technology, Bennett University, Greater Noida, India
[2] University School of Automation and Robotics, Guru Gobind Singh Indraprastha University, Delhi, India
[3] School of Information Communication and Technology, Bahrain Polytechnic, Isa Town, Bahrain
[4] Department of Information Technology, College of Computers and Information Technology, Taif University, Taif, Saudi Arabia



Corresponding author
Mamoon Rashid,
mamoon873@gmail.com

## ABSTRACT

**Background:** Lung cancer has the highest global fatality rate, with diagnosis primarily relying on histological tissue sample analysis. Accurate classification is critical for treatment planning and patient outcomes.

**Methods:** This study develops a computer-assisted diagnosis system for non-small cell lung cancer histology classification, utilizing the FastAI-2 framework with a modified ResNet-34 architecture. The methodology includes stain normalization using LAB colour space for colour consistency, followed by deep learning-based classification. The proposed model is trained on the LC25000 dataset and compared with VGG11 and SqueezeNet1_1, demonstrating modified ResNet-34's optimal balance between depth and performance. FastAI-2 enhances computational efficiency, enabling rapid convergence with minimal training time.

**Results:** The proposed system achieved 99.78% accuracy, confirming the effectiveness of automated lung cancer histopathology classification. This study highlights the potential of artificial intelligence (AI)-driven diagnostic tools to assist pathologists by improving accuracy, reducing workload, and enhancing decision-making in clinical settings.

## INTRODUCTION

Worldwide, there were an estimated 20 million new cancer diagnoses and 10 million cancer deaths. The burden is estimated to grow by almost 60% in the next 20 years, and by 2040, there could be 30 million new cases, with the greatest increase in low- and middle-income nations. In the western world, the incidence of cancer is likely to rise by 57% with an estimated 6.23 million cases by 2040. Of the present 20 million cases, 11 million are male with six million deaths and 10 million are female with five million deaths (*Siegel et al., 2023*; *American Cancer Society*). Lung cancer is the second most prevalent

cancer in both men and women, and a major cause of cancer deaths globally, accounting for 5.9% of diagnoses and 8.1% of cancer deaths (*American Cancer Society*). In India, lung cancer is the single largest cause of cancer death, largely because of excessive tobacco consumption and smoking. Lung cancer is divided into two broad categories: small cell lung cancer and non-small cell lung cancer (NSCLC), which are differentiated based on their histological features. NSCLC is divided into adenocarcinoma, squamous cell carcinoma, and large cell carcinoma, whereas small cell lung cancer is divided into small cell carcinoma and combined small cell carcinoma (*Hanna et al., 2017*).

This research involves the classification of NSCLC histology into three groups: adenocarcinoma, squamous cell carcinoma, and benign lung tissue, using a new method that combines color normalization with FastAI-2 and a ResNet-34 modified architecture. Figure 1 displays the classification standards for lung histology.

It is well known that patients diagnosed with adenocarcinoma tend to have a worse prognosis than those diagnosed with squamous cell carcinoma. However, the survival rate for patients with squamous cell carcinoma is comparatively lower than for those with adenocarcinoma (*Kawase et al., 2011*). This discrepancy is likely influenced by smoking-related comorbidities, which are more prevalent among squamous cell carcinoma cases, contributing to reduced survival rates. Despite these observations, a complete understanding of the differential biological aggressiveness of these two subtypes remains elusive.

Epidemiological studies indicate that adenocarcinoma is more common in women, whereas squamous cell carcinoma accounts for 10–30% of cases in women and 30–55% in men (*Hanna et al., 2017*; *Patel, 2005*; *PathologyOutlines.com, 2023*). The prevalence of these subtypes suggests potential gender-related factors in disease development, warranting further investigation into their underlying pathophysiology.

In modern healthcare diagnostics, medical practitioners primarily rely on medical imaging technologies for disease analysis. Among these, biopsy-based histological examination remains the most reliable diagnostic approach, involving the extraction of tissue samples for microscopic evaluation. Clinical medicine and laboratory research heavily depend on this technique (*Solis et al., 2012*). However, manual preparation of pathology slides is a complex and meticulous process, requiring precision and expertise. Additionally, the subsequent diagnostic analysis is time-intensive, necessitating skilled pathologists to examine lung cancer tissues under various magnifications (*Wei et al., 2019*). Addressing these challenges is essential to maximize the information obtained from pathology slides, ultimately ensuring accurate diagnosis and appropriate treatment administration.

Manually counting the numerous biological characteristics is a time-consuming and inefficient process for pathologists at this point. In addition, there is a potential that crucial characteristics may be missed due to overwork, which may result in an inaccurate diagnosis. This could lead to an incorrect treatment plan. Therefore computer-assisted classification of lung cancer is a prerequisite for early detection, and timely treatment can

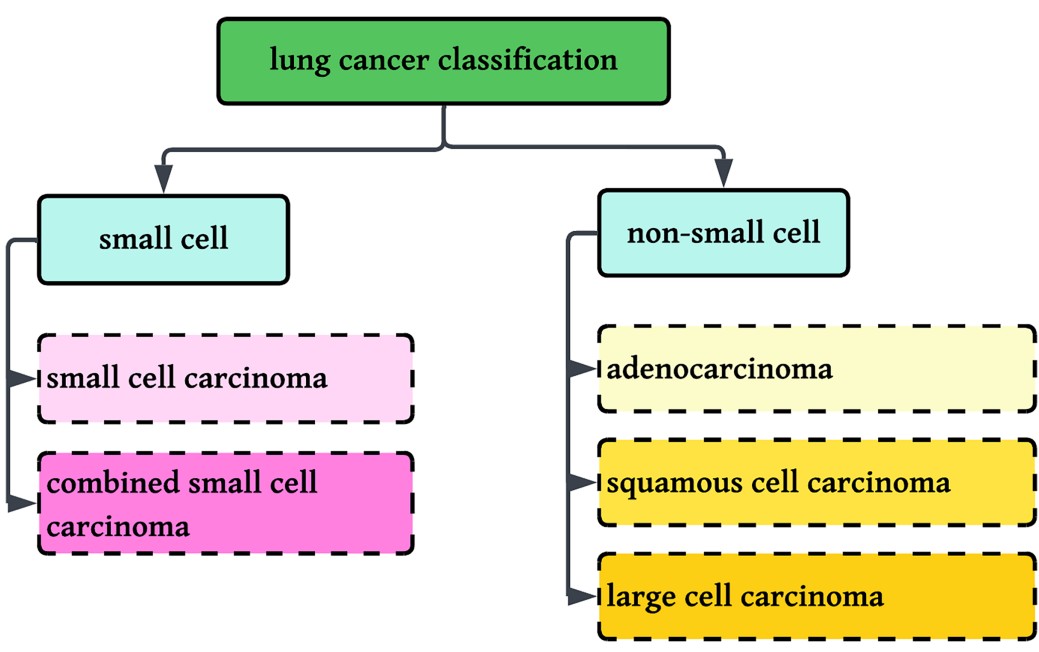

**Figure 1 Classification of Lung cancer into sub-categories.**

prolong the patient's life and chances of resubmission (*Saxena & Goyal, 2022*; *Gurcan et al., 2009*). To accurately identify lung cancer in the current study, the technology known as FastAI-2 is combined with the modified ResNet-34 model is used. Compared to its other possible alternatives, modified ResNet-34 has fewer layers but approximately the same performance level with less time. Deep learning results can be approximated more quickly with the help of FastAI-2. Enhancing the efficiency of learning models by leveraging GPU acceleration and optimizing the callback mechanism can potentially expedite model execution, reduce code complexity, and consequently enhance the accuracy of histopathology slide identification for lung cancer (*Praveen et al., 2022*). The Residual Network (ResNet) has been empirically demonstrated to effectively address the issue of vanishing gradients and facilitate efficient feature learning.

In contrast to past research that used raw histopathological images exclusively, this study presents a new method by coupling color normalization with FastAI-2 and a modified ResNet-34 network for enhanced multi-class lung cancer histopathology classification. The new method classifies three types: Adenocarcinoma, Squamous Cell Carcinina, and Normal. Through stain normalization, the new method corrects color variability challenges inherent in histopathological images, providing robust and consistent feature extraction. In addition, FastAI-2 also boosts computational efficiency by applying the one-cycle learning rate policy, which adaptively varies learning rates throughout training to ensure quicker convergence without affecting accuracy (*Smith, 2018*). The method saves training time by improving the learning process. Moreover, the

framework exploits transfer learning with pre-trained ResNet-34 models to enable fast model adaptation and reduce training time (*Howard & Gugger, 2020*). The comparative analysis with other deep learning architectures, such as VGG11 and SqueezeNet1_1, proves the superiority of the suggested model with an overall accuracy of 99.78% and much less execution time. The performance of the model is assessed using multi-class metrics such as macro, micro, and weighted averages for precision, recall, and F1-score to provide a complete assessment. The integration of model optimization, color normalization, and effective training methodologies in FastAI-2 enhances the automated lung cancer classification by improving accuracy, minimizing computational expense, and achieving faster convergence. These results portend the clinical utility of the model in assisting healthcare practitioners with accurate diagnostic lung cancer, which will in turn lead to enhanced patient care.

This work showcases a new solution to multi-class lung cancer histology classification that proves to work well for three classes. Summarily, the contributions in this article include:

- Applied color normalization for consistent and robust extraction of features.
- Improved FastAI-2 model with a modified ResNet-34 architecture to achieve efficient classification.
- Evaluated the performance of the model employing multi-class assessment metrics and shown to outperform VGG11 and SqueezeNet1_1.
- Enhanced computational efficacy with one-cycle learning rate strategy and transfer learning.

The subsequent sections are organized in the following manner. "Related Work" reviews existing techniques to classify lung cancer based on histopathological data. The proposed methodology is presented in "Proposed Methodology". The result outcomes derived from evaluating the proposed methodology are elaborated in "Result Analysis and Discussion". "Conclusion" provides the conclusion of the article.

## RELATED WORK

This article critically reviews different types of datasets used in the classification process of lung histology. Overall classification approaches are discussed in Table 1 based on three fundamental criterions. In the first criteria, different feature extraction techniques are reviewed. These techniques are responsible to construct the feature vectors. In the second criteria, various classifications approaches are reviewed to classify lung histology into respective classes. Finally, various performance evaluation criterion is compared. This literature is only confined to histopathological images. Other types of images (like computed tomography (CT), high-resolution computed tomography (HRCT)) are not part of this article.

Based on the review, it can be concluded that two public (*Al-Jabbar et al., 2023*; *Liu et al., 2022*; *Ali & Ali, 2021*; *Civit-Masot et al., 2022*; *Adu et al., 2021*;

**Table 1 Summary of related works with their findings.**

| Ref | Dataset | Feature extraction techniques | Classification approach | Performance evaluation |
|---|---|---|---|---|
| *Al-Jabbar et al. (2023)* | LC25000, Histopathological images, high resolution | Method 1: feature extraction from GoogLeNet, VGG-19 separately.<br>Method 2: Combined features extracted from GoogLeNet, VGG-19, followed by PCA for feature optimality.<br>Method 3: Fusion features from CNN models followed by hand-crafted features. | Artificial neural network (ANN) is used for classification. | Method 1: accuracy for GoogLeNet and VGG-19 are 95.5% and 95.9%, respectively.<br>Method 2: Mixed features after PCA are 98.7, and mixed features before PCA 98.5 in method-3 accuracy is 99.6%. |
| *Liu et al. (2022)* | 766 cases were recorded from 2019 to 2020 at the First Hospital of Baiqiu'en. The dataset was obtained at 20X whole slide images (WSI). | Feature extraction based on visual activation function and CroReLU additionally a priori knowledge of pathology | Deep learning models (SENet50 +CroReLU & MobileNet_CroReLU) | The diagnosis accuracy is reached up to 98.35%. |
| *Ali & Ali (2021)* | LC25000, histopathological images | Convolutional Neural Network | Multi-input capsule network | 99.58% of accuracy |
| *Civit-Masot et al. (2022)* | LC25000, Histopathological Images | Deep Neural Network | Explainable Deep Neural Network | System accuracy varies between 97.11% to 99.69% |
| *Adu et al. (2021)* | LC25000, Histopathological Images | The encoder feature fusion aggregates to extract features from the 2-lane convolutional neural layers. | Classification is done based on the new horizontal squash capsule network abbreviated as (DHS-CapsNet) to classify lung and colon cancers. | The system can generate 99.23% of accuracy. |
| *Mangal, Chaurasia & Khajanchi (2020)* | LC25000, Histopathological images | CNN is used to extract the features | These CNN features are classified by fully connected layers of deep application | 97.92 training accuracy is calculated, followed by 97.89% of validation accuracy. |
| *Talukder et al. (2022)* | LC25000, Histopathological images | Features are extracted by different transfer learning deep networks (VGG16, VGG19, MobileNet, DenseNet169, DenseNet201) | Extracted features from transfer learning approaches are fed into different machine learning approaches (RG, LR, XGB, SVM, MLP, LGB) followed by an ensemble to evaluate the performance | 99.05% of accuracy is reported while classifying lung cancer histology |
| *Mehmood et al. (2022)* | LC25000, Histopathological images | Transfer Learning techniques are used for feature extraction. | Modified AlexNet with class-selective image processing techniques | 98.4% of accuracy is achieved with this approach |
| *Toğaçar (2021)* | LC25000, Histopathological images | The feature set extracted from the DarkNet-19 network then run through the Equilibrium and Manta Ray Foraging optimization algorithms to select the inefficient features. | Then support vector machine (SVM) classifier is used to classify the Lung histology | SVM classifiers show a verypromising 99.69% of accuracy. |
| *Masud et al. (2021)* | LC25000, Histopathological images | Digital image processing techniques (2D Fourier features, 2D wavelet features) are | CNN is used to classify Lung histology | The classification accuracy of the proposed system is 96.33%. |

*(Continued)*

| Ref | Dataset | Feature extraction techniques | Classification approach | Performance evaluation |
| --- | --- | --- | --- | --- |
| *Nishio et al. (2021)* | Database 1: private dataset containing 94 images with 1,600× 1,200 pixel with RGB channel at 100X resolution. Dataset 2: LC25000, Histopathological Images (public dataset) | Two types of features extraction techniques are used. Method 1: Based on conventional texture analysis. Method 2: Homology-based image processing techniques. | Eight machine learning classifiers are used (perceptron model, logistic regression (LR), k-nearest neighbor (kNN), SVM with radial basis function (RBF), decision tree (DT), Random Forest (RF), gradient tree boosting (GTB)). | Maximum accuracy of 99.4% is recorded among all the classifiers. |
| *Hamida et al. (2021)* | AiCOLO colon cancer dataset | ImageNet generates a rich collection of learnt features in order to compensate for the lack of abundant WSI datasets. | To ensure the patch-level classification of WSI is done using ResNets. | Testing and evaluation accuracy is achieved at 96.98% |
| *Coudray et al. (2018)* | Data base of whole-slide images collected from The Cancer Genome Atlas | Deep learning | Deep learning (Inception V3) | AUC is 0.97 is determined |

*Mangal, Chaurasia & Khajanchi, 2020*; *Talukder et al., 2022*; *Mehmood et al., 2022*; *Toğaçar, 2021*; *Masud et al., 2021*; *Nishio et al., 2021*) and two private (*Al-Jabbar et al., 2023*; *Hamida et al., 2021*; *Coudray et al., 2018*) datasets have been utilized for lung cancer classification studies. Furthermore, these datasets have been analyzed using various feature extraction techniques, where key quality parameters such as texture, shape, and intensity-based features have been derived to enhance classification performance. Some researchers use transfer learning techniques (*Al-Jabbar et al., 2023*; *Civit-Masot et al., 2022*; *Adu et al., 2021*; *Mehmood et al., 2022*; *Hamida et al., 2021*) and other uses transfer learning techniques with feature selection technique like principal component analysis (PCA) (*Al-Jabbar et al., 2023*), Mante ray foraging optimization algorithm (*Masud et al., 2021*). Feature extraction techniques like fusion features from the convolutional neural network (CNN) model followed by hand-crafted features (*Al-Jabbar et al., 2023*), visual activation function (AF) with cross-ReLU along with prior knowledge of pathology (*Liu et al., 2022*), convolutional neural network (*Ali & Ali, 2021*; *Mangal, Chaurasia & Khajanchi, 2020*), CNN followed by encoder feature fusion as aggregators (*Adu et al., 2021*), and digital image procedures (*Masud et al., 2021*; *Nishio et al., 2021*) are also used to extract the features. These extracted features are fed into different classifiers to classify lung histology. Some researchers use machine learning techniques (*Al-Jabbar et al., 2023*; *Talukder et al., 2022*; *Toğaçar, 2021*; *Nishio et al., 2021*), and other uses deep networks (*Liu et al., 2022*; *Civit-Masot et al., 2022*; *Mangal, Chaurasia & Khajanchi, 2020*; *Mehmood et al., 2022*; *Masud et al., 2021*; *Hamida et al., 2021*) followed by multi-input capsule network (*Ali & Ali, 2021*; *Adu et al., 2021*). Classification accuracy among all the classifiers is compared and recorded. Accuracies among all the classifiers vary from a minimum of 95.5% to a maximum of 99.6%. This article classifies Lung histology using proposed colour normalization followed by fastai2 with a modified ResNet34 network.

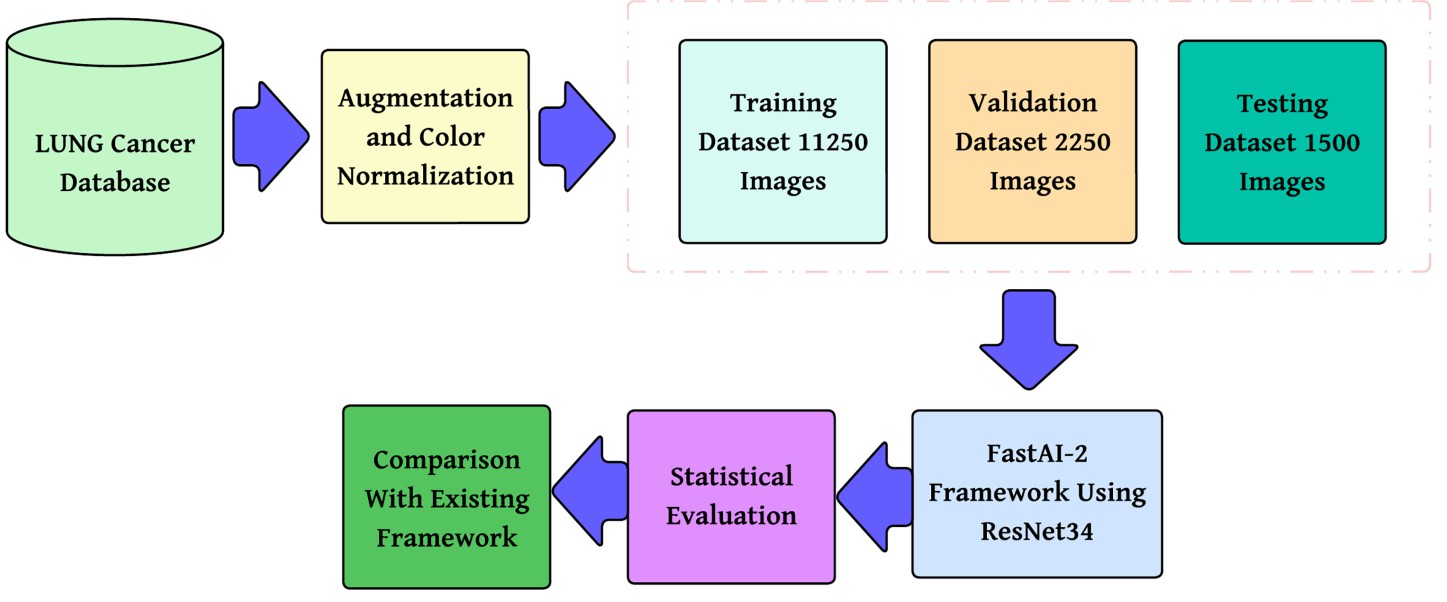

**Figure 2  Methodology of the proposed system.**

## PROPOSED METHODOLOGY

The methodology proposed, as shown in Fig. 2, is a step-by-step multi-class lung cancer classification following a structured approach. It starts with color normalization, which improves image quality and minimizes stain variability. The processed dataset is divided into training, validation, and testing subsets to provide reliable model evaluation. These pre-processed images are then passed through the FastAI-2 framework, where they are subjected to tensor transformation, feature extraction with a ResNet-34 model modified for this purpose, and classification by a fully connected Softmax layer into three classes: Adenocarcinoma, Squamous Cell Carcinoma, and Normal. The performance of the model is measured in terms of multi-class evaluation metrics such as overall accuracy, macro, micro, and weighted averages for precision, recall, and F1-score. Moreover, a confusion matrix is utilized to evaluate class-wise predictions. The performance measures are compared with state-of-the-art frameworks to ensure the efficacy of the model and prove its superiority in multi-class lung cancer classification.

### Dataset description

The LC25000 dataset, proposed by *Borkowski et al. (2019)*, is an open-source histopathological image dataset aimed at facilitating machine learning research for cancer diagnosis. It contains 25,000 colour images, divided equally into five different categories, such as lung adenocarcinoma, lung squamous cell carcinoma, benign lung tissue, colon adenocarcinoma, and benign colon tissue. Each image was originally taken from histopathological slides at a 1,024 × 768 pixel resolution and then resized to 768 × 768 pixels to keep an equal aspect ratio appropriate for deep learning tasks.

Here, we address the issue of lung cancer histopathology classification alone, *i.e.*, the classification of images into three categories: lung adenocarcinoma, lung squamous cell carcinoma, and benign lung tissue. The data were already preprocessed during the time of data acquisition and augmented to eliminate additional preprocessing requirements of augmentation. The initial augmentation step was carried out with the Augmentor software tool (*Borkowski et al., 2019*), implementing random left and right rotation (max 25 degrees, probability = 1.0) and horizontal and vertical flip (probability = 0.5) to provide a rich representation of histopathological variations. The pre-augmented dataset has been tested to be HIPAA-compliant, maintaining patient confidentiality, and forms the standard utility for deep learning model development in cancer classification. With the use of this dataset, scientists can investigate strong machine learning methods for separating malignant from benign lung tissue that can lead to improvements in AI-based pathology diagnostics (*Borkowski et al., 2019*).

## Colour-normalizing of the histopathological image from the LC25000 dataset

Pre-processing is carried out to enhance images by suppressing noise and enhancing crucial characteristics. As a result, crucial information is extracted from the images, and the images become compatible with deep-learning and machine-learning networks. Noise can be seen in the images of the LC25000 dataset because the biopsy was combined with a variety of medical materials, and there is not enough contrast between the afflicted tissue and the surrounding tissue. This article explains a technique for a broader type of colour correction known as colour borrowing, in which the colour properties of one image are taken from another image. The following series of steps are followed for this process. In the First step, the augmented image is converted into $L^*a^*b^*$ colour space, followed by $L^*a^*b^*$ colour space, which is projected on decorrelated colour space using the borrower image. Then finally, the projected image is back to RGB colour space from the $L^*a^*b^*$ colour space.

**Step 1: Conversion of augmented RGB image into $L^*a^*b^*$ colour-space**

In the RGB colour space, if the blue channel is dominant, it means that the blue intensity is significantly higher relative to the red and green channels. This affects the overall perception of colour, requiring adjustments across all three channels to ensure a consistent colour transformation. Since RGB channels are interdependent, modifying one component without adjusting the others can lead to unnatural colour shifts, making simultaneous adjustments necessary for maintaining colour consistency. Any technique that modifies colours is made more difficult due to this. A colour space that is orthogonal and free of connections between its axes is precisely what we are looking. $L^*a^*b$ colour space can be the answer to this. During the conversion from RGB to $L^*a^*b$ colour model, the Euclidean distance is calculated since the difference in perceived value between two colours in the $L^*a^*b^*$ colour space is consistent across all observers.

Three different aspects make up the $L^*a^*b^*$ colour space: the luminance component $L^*$, and the chromaticity components $a^*$ and $b^*$. These components represent the position of colour along the red-green and blue-yellow axes, respectively (*Solis et al., 2012*). The
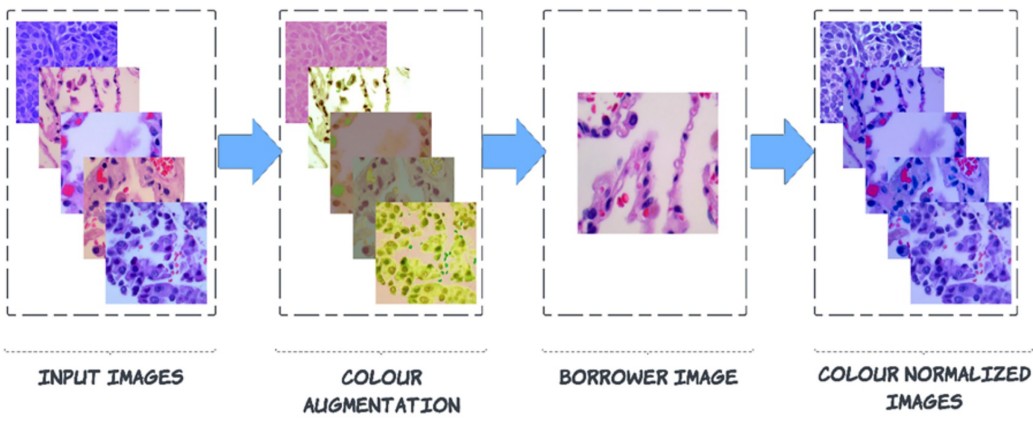

**Figure 3** Steps to colour normalization.

conversion formula initially establishes the tri-stimulus coefficients in the following manner:

$$X = 0.43R + 0.35G + 0.19B$$
$$Y = 0.23R + 0.71G + 0.07B \tag{1}$$
$$Z = 0.02R + 0.13G + 0.94B.$$

The calculation of the CIELab colour model is as follows:

$$L^* = 116\left\{h\left(\frac{Y}{Y_s}\right)\right\} - 16$$
$$a^* = 500\left\{h\left(\frac{X}{X_s}\right)\right\} - h\left(\frac{Y}{Y_s}\right) \tag{2}$$
$$b^* = 200\left\{h\left(\frac{Y}{Y_s}\right)\right\} - h\left(\frac{Z}{Z_s}\right)$$

where the usual stimulus coefficients $X_s$, $Y_s$, and $Z_s$ represent the reference white values in the CIE standard illuminant, which are used to normalize the colour transformation process for ensuring perceptual consistency. After calculating L*a*b channels, the de-correlation of colour space is started based on the borrower image (discussed in Fig. 3).

**Step 2: De-correlation of colour space using borrower image**

Consider an image $I(i, j, k)$, where $i$, $j$ are the height and width of the image, respectively, and $k$ represents the number of channels of the image. Then Eq. (3) decorrelates the pixel's intensity.

$$Inp_{img}[i, j, k] = \left[\left(Inp_{img} - Inp_{img}(mean[k])\right) * \left\{\frac{Bor_{img}(std[k])}{Inp_{img}(std[k])}\right\}\right] + Bor_{img}(mean[k]). \tag{3}$$

$Inp_{img}$ is the colour-augmented input image, $Inp_{img}(mean[k])$ represents the mean of the inputted image of each channel, $Bor_{img}(std[k])$ keeps a record of the standard deviation of borrower image and $Inp_{img}(std[k])$ handle standard deviation of inputted image. $Bor_{img}(mean[k])$ will keep track of the mean of borrower images for each channel. After

these colour processing adjustments, the next step is again back to RGB colour space from L*a*b* colour space (*Reinhard et al., 2001*; *Shen et al., 2022*).

**Step 3: Image back to RGB from L\*a\*b\* colour space**

An inverse procedure converts L*a*b* channels into RGB colour space. Equations. (1) and (2) are redefined as follows:

$$\begin{bmatrix} X \\ Y \\ Z \end{bmatrix} = \begin{bmatrix} 1 & 1 & 1 \\ 1 & 1 & -1 \\ 1 & -2 & 0 \end{bmatrix} \cdot \begin{bmatrix} 0.58 & 0 & 0 \\ 0 & 0.41 & 0 \\ 0 & 0 & 0.71 \end{bmatrix} \cdot \begin{bmatrix} l \\ a \\ b \end{bmatrix}. \tag{4}$$

Equation (4), *LAB* channels are converted into *XYZ* stimulus using matrix multiplication. After that, we can transform the data from the *XYZ* stimulus into the *RGB* colour space by raising the pixel values to the power of ten to return to linear space. Equation (5) helps to understand the procedure.

$$\begin{bmatrix} R \\ G \\ B \end{bmatrix} = \begin{bmatrix} 4.47 & -3.59 & 0.12 \\ -1.2 & 2.4 & -0.17 \\ 0.05 & -0.25 & 1.21 \end{bmatrix} \cdot \begin{bmatrix} X \\ Y \\ Z \end{bmatrix} \tag{5}$$

After removing all the anomalies, these colour-normalized images are fed into deep learning frames to work for further classification of lung histology.

## Classification framework

FastAI-2 is a deep learning framework developed on top of PyTorch to simplify model development while offering room for sophisticated customization. It incorporates several machine learning libraries and tools to enable researchers to obtain efficient training and precise results with little code. Unlike other deep learning frameworks that involve much configuration, FastAI-2 offers high-level APIs to enable quick experimentation while still allowing access to low-level PyTorch functionality to fine-tune model performance.

The structure of the framework follows three core design principles: rapid productivity, simple configuration, and adaptable architecture. It is realized through modular and layered APIs so that both newcomers and deep learning experts can make use of it. It further provides automatic hyperparameter search, learning rate scaling, and data augmentation with its built-in functions, promoting more efficient training. By linking high-level convenience and low-level personalization, FastAI-2 is a capable tool for applying deep learning across different areas (*Praveen et al., 2022*).

Two of the most important design goals that FastAI-2 strives to achieve are accessibility and productivity while maintaining flexibility. The residual neural network, often known as ResNet-34, is a 34-layer. In the foundation of modified ResNet34, we need to define key operations, including convolutional layers, batch normalization, ReLU activation, residual connections, pooling, and fully connected layers. Below is a structured approach to formulating the mathematical backbone of your modified ResNet-34.

*Convolutional layer*: Each convolutional layer applies a set of learnable filters $W$ to the input tensor $X$, producing a feature map $Y$. The convolution operation is given by:

$$Y_{i,j,k} = \sum_m \sum_n \sum_c W_{m,n,c,k} . X_{i+m,j+n,\,c} + b_k \tag{6}$$

where, in Eq. (6), $X_{i+m,j+n,\,c}$ is the input pixel at position $(i+m,\ j+n)$ in channel $c$, $W_{m,n,c,k}$ is the weight of the filter for kernel size $(m, n)$, $b_k$ is the bias term, $Y_{i,j,k}$ is the output feature map at position $(i, j)$ for channel $k$. In modified ResNet-34, these convolutional layers are followed by batch normalization and ReLU activation.

*Batch normalization:* Batch normalization normalizes activations before passing them to the next layer. Given an activation $a$, the normalized output is $\hat{a} = {}^{a\,-\,\mu}\!/_{\sigma}$ where $\mu$ is the mean of activations and $\sigma$ is the standard deviation. Batch normalization introduces learnable parameters $\gamma$ (scale) and $\beta$ (shift), giving the final transformation *Batch normalization* $=\ \gamma\hat{a} +\ \beta$.

*ReLU activation:* ReLU is applied element-wise $f(a) = max\ (0,\ a)$. This function preserved non-linearity while preserving positive activation. The modified ResNet34 residual learning, where the output of a block is computed as, $Y = f(X) + X$, where $f(X)$ represents a sequence of convolutional layers, $X$ is the input that is directly added to the output through the skip connection. This helps mitigate the vanishing gradient problem, allowing deeper networks to learn effectively.

*Adaptive average pooling:* Instead of a fixed pooling size, Adaptive Average Pooling reduces the feature map dynamically to a fixed-size tensor. Given an input tensor $X$ of shape $(H,\ W)$, the pooled output $Y$ is computed as:

$$Y_{i,j} =\ \frac{1}{H'W'} \sum_{p=1}^{H'} \sum_{q=1}^{W'} X_{i+p,\,j+q}. \tag{7}$$

This ensures a consistent feature map size for the fully connected layer, regardless of input image dimensions. In Eq. (7), $Y_{i,j}$ is the output of the adaptive average pooling operation at position $(i, j)$ in the pooled feature map. $X_{i+p,\,j+q}$ is the input feature map value at position $(i+p,\ j+q)$, which is part of the receptive field being averaged. $H'$ *is* height of the pooling region (window size in the vertical direction). $W'$ is the width of the pooling region (window size in the horizontal direction). Adaptive Average Pooling computes the average value of a local region of size $H' \times W'$ in the input feature map and assigns the result to a single pixel in the output feature map. The adaptive nature ensures that the feature map is resized dynamically based on the required output dimensions.

*Fully connected layer and SoftMax activation:* The final classification layer maps the feature vector FFF to output probabilities $z = WF + b$, where, $W$ is the weight matrix, $F$ is the flattened feature vector, $b$ is the bias term, and $z$ is the logits vector before applying Softmax. In this article $P(y_i) = \dfrac{e^{z_i}}{\sum_j e^{z_i}}$ is used as the Softmax function, where $P(y_i)$ is the probability of class $i$, ensuring that all outputs sum to 1. This foundation describes how modified ResNet-34 operates, incorporating convolutional layers, batch normalization, residual learning, adaptive pooling, and Softmax classification. These modifications ensure robust feature extraction, efficient gradient flow, and optimized classification for lung cancer histopathology. A modified ResNet-34 architecture consisting of $r$ residual blocks would provide $2^r$ several potential pathways for processing the data. This is because each residual block in ResNet would create two different paths. As a direct result, decreasing the

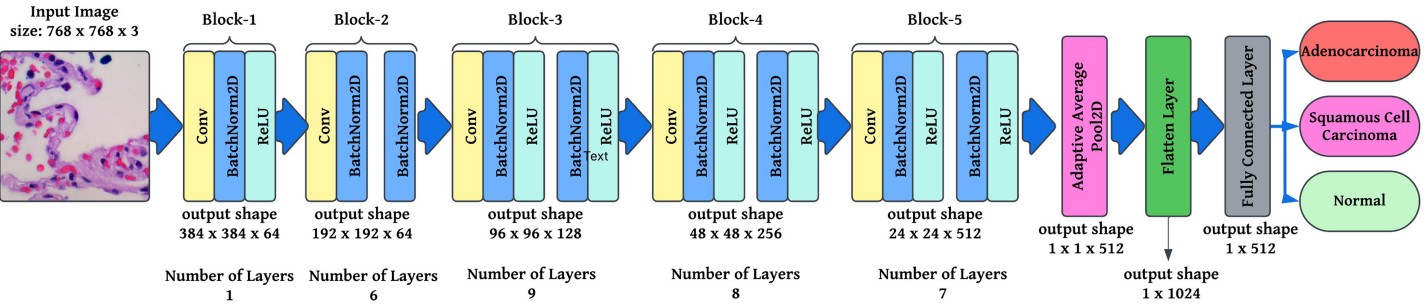

**Figure 4** Semantic representation of ResNet-34 architecture with layering details.

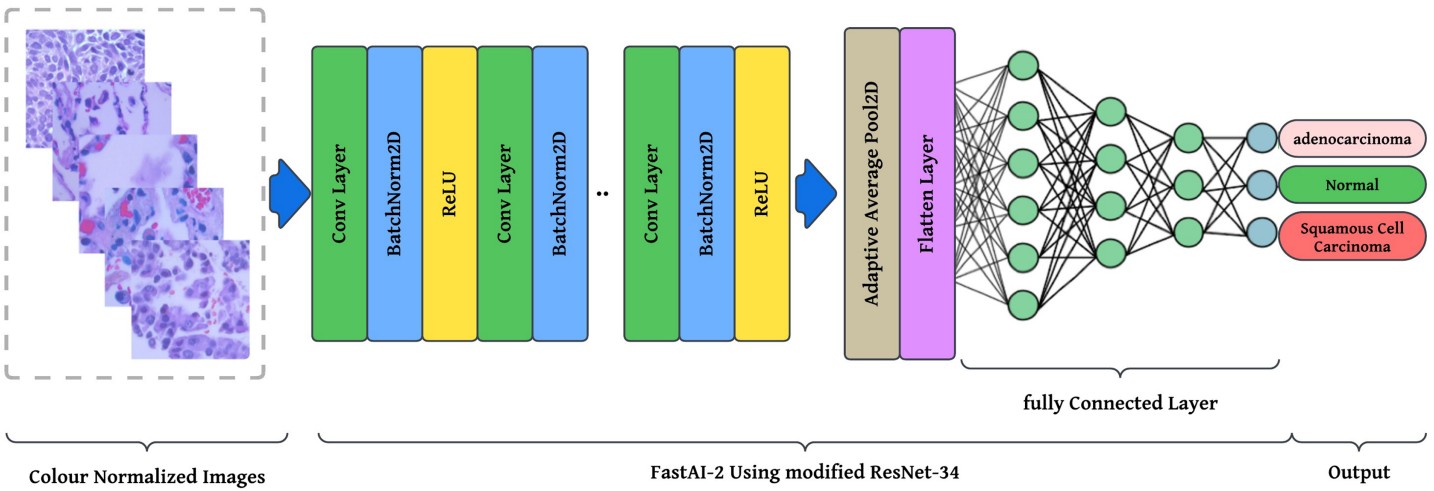

**Figure 5** Graphical representation of FastAI-2 architecture with the pre-processed input image.

total number of layers in the design will not significantly impact the efficiency with which it functions. Working with a lower number of layers would result in faster computations and an improvement in the capability of training networks. Figure 4 illustrates the layered approach the ResNet-34 model takes to solve the problem.

The preprocessing stage involves resizing images to 768 × 768 pixels to maintain a uniform aspect ratio, followed by colour normalization using Lab colour space transformation with decorrelation to ensure stain consistency across histopathological slides. These preprocessed images are then fed into the modified ResNet-34 model with FastAI-2 (Fig. 5) for classification into three categories: Normal, Adenocarcinoma, and Squamous Cell Carcinoma. The performance of the proposed system is evaluated and compared against existing approaches to demonstrate its effectiveness. The article illustrates the processing pipeline of the modified ResNet-34 model used in this study. The input consists of colour-normalized histopathological images, which undergo multiple stages of feature extraction, pooling, and classification within the FastAI-2 framework utilizing modified ResNet-34.

**Table 2 Distribution of a number of images in each category along with dispersal among train_set (75%), validation_set (15%), and test_set (10%).**

| Class of lung cancer | train_set | validation_set | test_set | Total |
|---|---|---|---|---|
| lung_aca | 3,750 | 750 | 500 | 5,000 |
| lung_n | 3,750 | 750 | 500 | 5,000 |
| lung_scc | 3,750 | 750 | 500 | 5,000 |
| **Total** | **11,250** | **2,250** | **1,500** | **15,000** |

This visualization effectively demonstrates how the proposed modifications, such as colour normalization, optimized convolutional blocks, and adaptive pooling, contribute to enhancing the model's robustness, efficiency, and classification accuracy. By integrating these components, the modified ResNet-34 ensures a computationally efficient and high-performance deep learning framework for lung cancer histopathology classification.

## RESULT ANALYSIS AND DISCUSSION

Google Colab has standard libraries on Windows 11 Home (version 21H2), the 64-bit operating system with Ryzen 5 3500 U AMD processor with Vega mobile Gfx. Moreover, 8 GB of RAM is used to classify lung histopathological images. The distribution of the database into tagged classes is displayed in Table 2.

The diagnostic performance of the proposed system is evaluated using multi-class performance measures, including overall accuracy, macro and micro precision, recall, and F1-score. In multi-class classification, overall accuracy is calculated as the proportion of correctly predicted instances out of the total number of samples, expressed as a percentage. Precision is evaluated using macro and micro averages, where macro precision calculates the precision for each class individually and then averages them, while micro precision aggregates contributions from all classes for a global precision score. Recall is also calculated using macro and micro averaging. Macro recall averages the recall values across all classes, whereas micro recall calculates the recall considering the sum of true positives and false negatives across all classes. The F1-score, a harmonic mean of precision and recall, is calculated using both macro and micro averaging to provide a balanced view of the model's performance.

This comprehensive approach ensures that the evaluation metrics account for class imbalance and provide a detailed performance analysis for each category. By combining macro and micro averaging, the predictive performance of the model is effectively summarized, ensuring robust evaluation across all classes.

### Tunning the hyperparameters

Training a deep neural network (DNN) is a problem of global optimization. Among the most important hyperparameters to fine-tune in the process of DNN training is the learning rate. An ill-chosen learning rate significantly impacts model performance: a small learning rate might result in training that takes longer and converges slowly, and a large
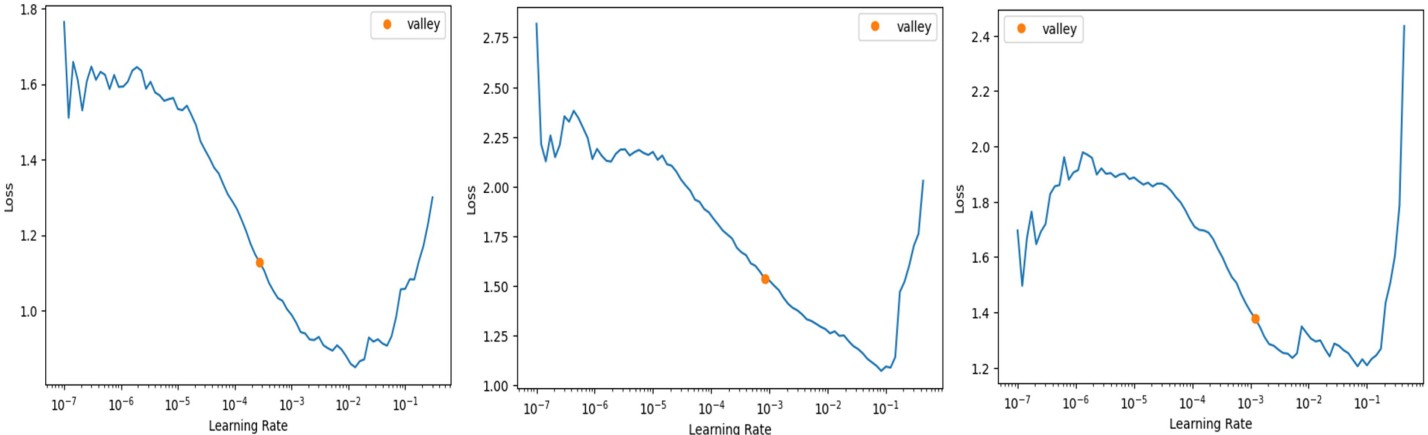

**Figure 6 Stimulation of optimal learning rate.** (A) ResNet34 network, (B) Vgg11 network, (C) SqueezeNet1_1 network.

learning rate might cause the loss function to oscillate or even diverge, rendering the model incapable of finding the optimal solution.

In FastAI-2, the learn.lr_find method is used to determine the optimal learning rate. This technique involves gradually increasing the learning rate after each mini-batch, while recording the corresponding loss function values. By plotting the losses against the learning rates, the behaviour of the loss function can be visualized, providing insights into the ideal learning rate for model training. Figure 6 illustrates the learning rate optimization process for three different architectures: modified ResNet-34, VGG-11, and SqueezeNet1_1. This approach enables the selection of a learning rate within the linear zone of rapid loss reduction, ensuring faster convergence and improved model performance. The use of FastAI-2's learning rate finder enhances the training efficiency for multi-class classification, as demonstrated across the tested networks.

The graph that is displayed above (Fig. 6) can be broken down into three distinct zones: the shallow zone, which describes an environment in which a change in learning rate has a negligible impact on loss, the linear zone, which describes an environment in which we observe a rapid decrease in the loss function, and the divergent zone, which describes an environment in which a learning rate that is too high causes loss to bounce and eventually diverge from the local minima. In a perfect world, we would conduct our job in the linear zone, characterized by the most significant decline in the loss function. We do not want to operate at the minimum because doing so would prevent us from being able to update the weights. The minimum is located at the point where the gradient of the loss function is not changing. A good rule of thumb would be to choose a location at least one magnitude higher than the absolute minimum. When we use *learn.lr_find*, we have the option of utilizing Valley to work just perfectly in our situation. The remaining parameters, such as the number of trainable parameters, are 0.55 million, epochs are 5, batch size is 32, the activation function is ReLU, the number of convolution layers is 31, the random state is 2, and two strides are taken in this experiment. 0.55 million trainable parameters were chosen

**Table 3 Analysis of training and validation loss during training for ResNet34 network.**

| Epoch | Train_loss | Valid_loss | Accuracy | Time |
| --- | --- | --- | --- | --- |
| 0 | 0.215012 | 0.087023 | 0.96652 | 10:28 |
| 1 | 0.082948 | 0.052266 | 0.984141 | 10:34 |
| 2 | 0.058602 | 0.033877 | 0.986784 | 10:32 |
| 3 | 0.018892 | 0.018289 | 0.994273 | 10:31 |
| 4 | 0.005173 | 0.005071 | 0.997797 | 10:32 |

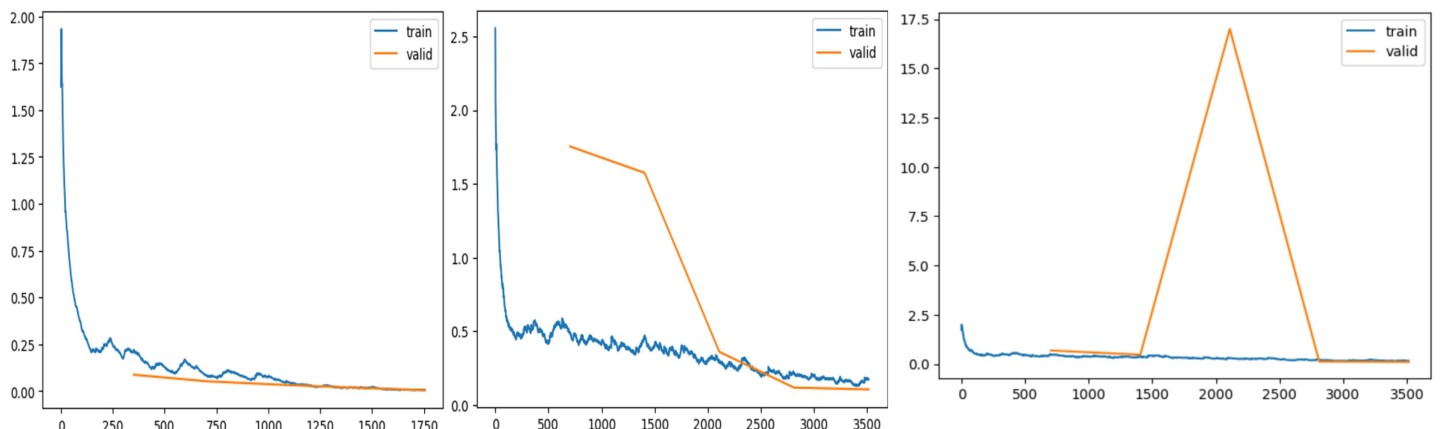

**Figure 7 Training and validation losses evaluation.** (A) ResNet34 network, (B) Vgg11 network, (C) SqueezeNet1_1 network.

to trade off model complexity and generalization, allowing efficient feature learning without overfitting. The epochs were fixed at five based on empirical experience using FastAI-2's one-cycle learning rate policy, where learning rates adapt dynamically to improve convergence (*Smith, 2018*). The model was able to achieve peak accuracy at five epochs after which no appreciable improvement was seen. A batch size of 32 was selected in order to achieve training stability and computational efficiency, and ReLU was employed because of its ease of use and capability to eliminate the vanishing gradient problem. The model employs 31 convolution layers for extraction of deep features to improve the accuracy of classification. Fixing the random state at 2 provides reproducibility of the results, whereas strides of 2 provide equilibrium between feature resolution extraction and efficiency in computation. These parameters were empirically adjusted to produce high accuracy and lower execution time, providing a strong and effective classification system.

## Training and validation losses
After tuning the hyper-parameters, the training and validation procedure is started, and for each epoch, training loss, validation loss, accuracy, and time taken in each epoch are recorded and displayed in Table 3.

**Table 4 Evaluation based on different performance measures.**

|  | Precision | Recall | F1-score | Accuracy |
|---|---|---|---|---|
| lung_aca | 1 | 0.99 | 1 | 0.9948 |
| lung_n | 1 | 1 | 1 | 1 |
| lung_scc | 0.99 | 1 | 1 | 0.9986 |

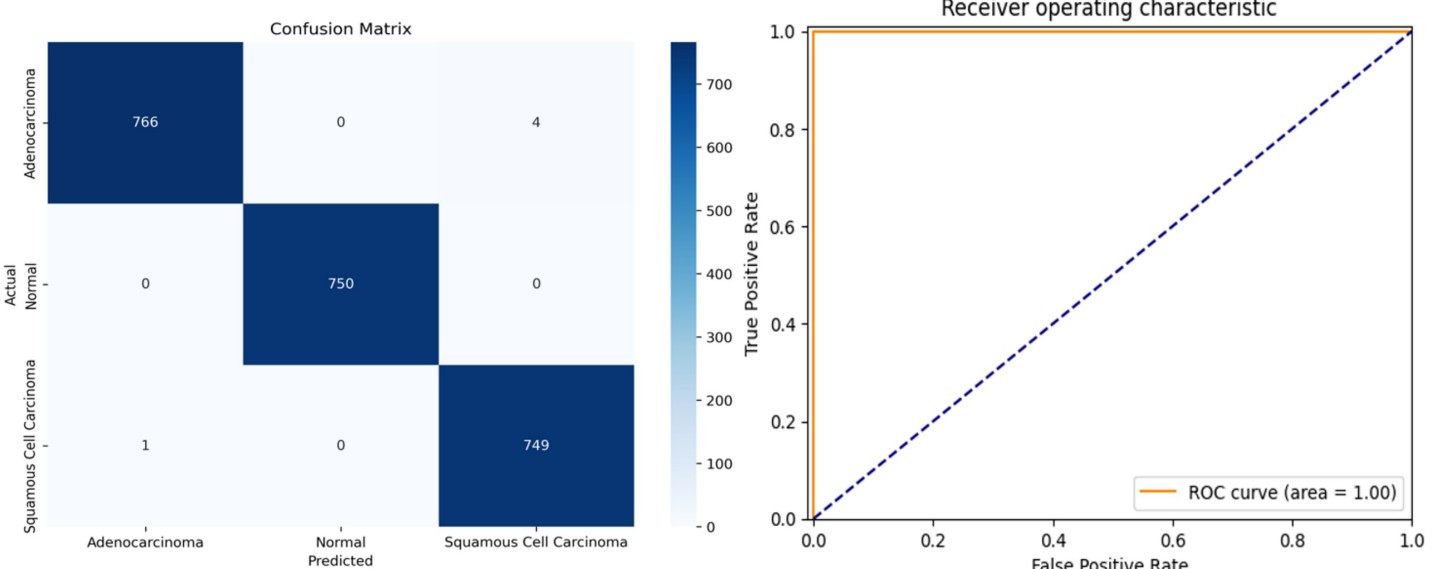

**Figure 8** (A) Confusion matrix and (B) receiver operating characteristic (ROC) curve of ResNet34 architecture.

**Table 5 Comparison based on performance among different pre-train networks.**

| Model | Precision (%) | Recall (%) | F1-score (%) | Accuracy (%) | ROC AUC (%) | Execution time (hh:mm:ss) |
|---|---|---|---|---|---|---|
| FastAI (ResNet-34) | 99.66 | 99.66 | 100 | 99.78 | 100 | 00:52:37 |
| FastAI (VGG-11) | 96.33 | 96.33 | 96 | 95.99 | 100 | 01:28:01 |
| FastAI (SqueezeNet1_1) | 96.67 | 95.67 | 95.33 | 95.59 | 100 | 00:19:18 |
| TensorFlow (ResNet-34) | 97.42 | 96.35 | 96.4 | 96.45 | 100 | 01:05:22 |
| TensorFlow (VGG-11) | 95.88 | 95.79 | 95.8 | 95.83 | 100 | 01:41:56 |
| TensorFlow (SqueezeNet1_1) | 96.1 | 95.12 | 95.5 | 95.42 | 100 | 00:25:43 |

Figure 7 shows validation and training losses during the procedure. Table 3 shows that training loss starts from 0.215 and continuously dips up to 0.00517 as the number of epochs increases. This show model is trained perfectly with derived learning_rate.

## Evaluation of the proposed framework

The proposed framework was rigorously tested on simulated data to evaluate its performance on previously unseen instances. To ensure an objective evaluation, the test data were pre-processed and categorized consistently with the training and validation sets.

squamous cell carcinoma/
adenocarcinoma/2.94/0.95

squamous cell carcinoma/
adenocarcinoma/1.29/0.73

squamous cell carcinoma/
adenocarcinoma/1.16/0.69

squamous cell carcinoma/
adenocarcinoma/.70/0.50

adenocarcinoma/squamous
cell carcinoma/.69/0.50

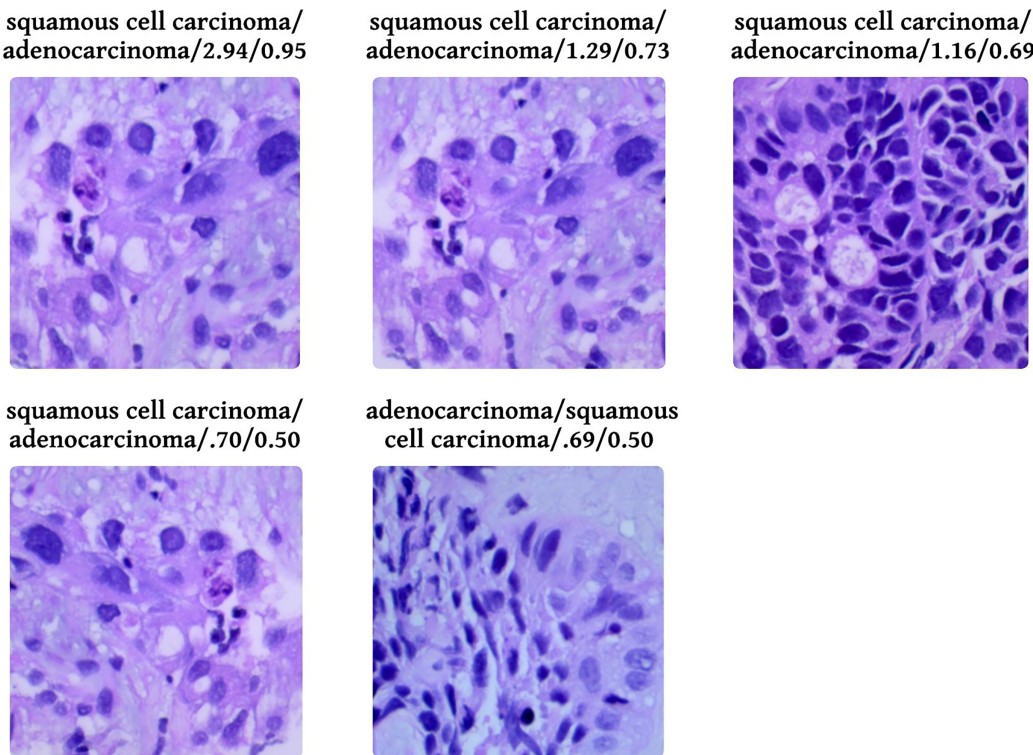

**Figure 9 Probabilistic analysis of loss between prediction and actual class.** Prediction/Actual/Loss/Probability.     

The multi-class performance metrics for the model's predictions on the test set are presented in Table 4, and the confusion matrix is illustrated in Fig. 8.

The overall accuracy of the model is 99.78%, as calculated using the Weighted Average approach. The model achieved a macro precision of 0.993, macro recall of 0.996, and macro F1-score of 0.996, demonstrating consistent performance across all classes. Similarly, the micro precision, recall, and F1-score were each 0.996, reflecting the model's strong predictive capability.

Class-specific accuracies are 99.48% for Adenocarcinoma, 99.86% for Squamous Cell Carcinoma, and 100% for Normal. These results highlight the model's ability to accurately distinguish between the three classes, achieving high precision, recall, and F1-scores for each category. The use of macro, micro, and weighted averages ensures a comprehensive evaluation, making the performance metrics more robust and reliable for multi-class classification.

Table 5 is a comparative study of deep learning models that were employed using FastAI (PyTorch) and TensorFlow to classify lung cancer histopathology. These models were compared based on their most important performance metrics, such as precision, recall, F1-score, accuracy, ROC AUC score, and run time. Of all the models, FastAI's ResNet-34 had the best accuracy of 99.78% with an F1-score of 100%, illustrating its high performance in classifying lung cancer histopathological images. In contrast, TensorFlow's ResNet-34

**Table 6 Comparison of the proposed model with the most recent existing literature.**

| References | Classification approaches | Accuracy |
|---|---|---|
| *PathologyOutlines.com (2023)* | GoogLeNet | 95.5% |
| | VGG-19 | 95.9% |
| | Mixed Feature before PCA | 98.6% |
| | Fusion Feature+hand crafted feature | 99.6% |
| *Solis et al. (2012)* | SENet50+CroReLU MobileNet+CroReLU | Up to 98.35% |
| *Gurcan et al. (2009)* | DHS-CapsNet | 99.23% |
| *Al-Jabbar et al. (2023)* | Ensemble machine learning approaches | 99.05% |
| *Liu et al. (2022)* | Transfer Learning (AlexNet) | |
| *Ali & Ali (2021)* | SVM | 99.69 |
| *Civit-Masot et al. (2022)* | CNN | 96.33 |
| *Mangal, Chaurasia & Khajanchi (2020)* | Machine Learning | 99.4 |
| **Proposed** | **FastAI-2 (ResNet34)** | **99.78%** |

achieved a slightly lower accuracy of 96.45%, indicating that FastAI's implementation optimizes model training more effectively.

When comparing VGG-11 models, FastAI's version achieved 95.99% accuracy, slightly outperforming TensorFlow's VGG-11 at 95.83%, though the latter took significantly longer to train (1 h 41 min *vs.* 1 h 28 min). SqueezeNet1_1, known for its computational efficiency, exhibited the fastest training times in both frameworks, with FastAI completing training in 00:19:18, compared to TensorFlow's 00:25:43. However, the trade-off for SqueezeNet1_1 was lower accuracy (95.59% in FastAI and 95.42% in TensorFlow), making it a viable option for resource-constrained environments but less suitable for high-precision classification tasks.

Overall, FastAI models consistently outperformed their TensorFlow counterparts in terms of accuracy and training efficiency, particularly in the case of modified ResNet-34, which achieved both higher classification performance and faster convergence in FastAI. While TensorFlow models required longer execution times, their accuracy remained close to that of FastAI, making them viable alternatives where framework flexibility or integration into existing TensorFlow-based pipelines is required. These findings highlight that FastAI provides a computational advantage with optimized learning techniques, making it a preferable choice for lung cancer histopathology classification.

Figure 9 shows five levelled images based on their predicted class by the proposed system, followed by an actual class of the image, classification loss, and probability of classification. Out of five images, two have a predicted probability of 50% with minimal loss; those are easily truncated by manual inspection. To provide evidence that the model that has been proposed is effective, we conduct a thorough analysis of the data that has been gathered and compare it to the findings that have been obtained using the most recent and cutting-edge methodologies. When the lesions of the lungs are classified concurrently, as shown in Table 6, the recommended model performs more effectively than the currently

thought of as being state-of-the-art methods. In terms of accuracy, the proposed model has surpassed the previously used one and is a better fit overall with the literature.

## CONCLUSION

In this article, we introduced a new uniform tone staining process combined with the FastAI-2 framework and modified ResNet-34 architecture to identify histopathological images of lung cancer into three groups: Adenocarcinoma, Squamous Cell Carcinoma, and Normal. The LC25000 dataset was used to train and test the model. For colour consistency and feature extraction improvement, a colour normalization method was used to ensure that all images are uniformly consistent. This preprocessing operation allowed the model to learn stronger features, leading to better classification accuracy.

The model proposed here scored a total accuracy of 99.78% and an F1-score of 100%, which was better than current models in the literature. Furthermore, incorporation of FastAI-2's learning rate finder enhanced model convergence, drastically shortening execution time. This method shows the feasibility of computer-aided diagnosis systems to aid pathologists in correctly diagnosing lung cancer cases with efficiency and lower costs.

Nevertheless, the current study depended mostly on the LC25000 dataset, and as such, it might not reflect the complete variety of histopathological images that occur in practice. The model's generalizability to other datasets and clinical situations thus cannot be tested. Additionally, the effects of staining variations and noise on the performance of the model were not examined. Although the suggested framework had high computational efficiency, its robustness and scalability on more extensive and complex datasets need further exploration.

Future work would involve testing the performance of the model on other varied datasets and checking for its flexibility to accommodate different staining methods. Performing ablation studies to examine the impact of each preprocessing step and checking the scalability of the model on larger datasets would give a better idea of the model's efficiency. Overcoming these limitations would enhance the use of the model in real-world diagnostic platforms and help push the field of artificial intelligence (AI) pathology diagnostics forward.

### Funding

This research was funded by Taif University, Saudi Arabia, Project No. (TU-DSPP-2024-52). The funders had no role in study design, data collection and analysis, decision to publish, or preparation of the manuscript.

### Grant Disclosures

The following grant information was disclosed by the authors:
Taif University, Saudi Arabia: TU-DSPP-2024-52.

## Competing Interests

The authors declare that they have no competing interests.

## Author Contributions

- Pranshu Saxena conceived and designed the experiments, performed the experiments, performed the computation work, prepared figures and/or tables, and approved the final draft.
- Sanjay Kumar Singh conceived and designed the experiments, performed the experiments, performed the computation work, prepared figures and/or tables, and approved the final draft.
- Mamoon Rashid analyzed the data, prepared figures and/or tables, and approved the final draft.
- Sultan S. Alshamrani analyzed the data, authored or reviewed drafts of the article, and approved the final draft.
- Mrim M. Alnfiai analyzed the data, authored or reviewed drafts of the article, and approved the final draft.

## Data Availability

The dataset are available at Zenodo:

- Borkowski, A. (2025). LC25000 [Data set]. Zenodo. https://doi.org/10.5281/zenodo.14998042.

The code are available at Zenodo:

- pranshusaxena149. (2025). pranshusaxena149/PeerJ-FastAI2: DOI:Code-FastAI2 (V0.0.2_FastAI2). Zenodo. https://doi.org/10.5281/zenodo.14997989.

## Supplemental Information

Supplemental information for this article can be found online at http://dx.doi.org/10.7717/peerj-cs.2903#supplemental-information.

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
