# Peer review of "Efficient deep learning model for classifying lung cancer images using normalized stain agnostic feature method and FastAI-2"

_PeerJ Computer Science, doi:10.7717/peerj-cs.2903_

## Round 0.1 · original submission · Major Revisions

I have decided that your manuscript requires Major Revisions before it can be considered for publication. While your study presents an interesting application of deep learning techniques, the reviewers have raised several critical concerns that must be addressed.

- The dataset labels should be removed from the Introduction section;
- The abbreviation lung_aca is not a standard term in medical practice. Please ensure that all labels are revised accordingly;
- There is an inconsistency in how your deep learning model is described. The manuscript first claims that a novel model was developed, but later states that ResNet-34 was used. Please clarify whether your study introduces a new model or applies an existing one;
- The methodology lacks sufficient detail regarding FastAI-2. The description should explicitly state how the framework is used in the classification process;
- The "Classification Framework" section requires additional technical details, particularly regarding (i) whether the model was pre-trained, (ii) the dataset used for pre-training (if applicable), and (iii) the structure and configuration of the classification layer;
- The study presents a multi-class classification problem, but binary classification metrics are used in the results section. This suggests a misalignment in evaluation methodology. Please ensure that appropriate multi-class performance metrics are reported;
- The unusually fast convergence (only five epochs) raises concerns. A more detailed justification of this result is needed, including an analysis of potential overfitting or underfitting issues;
- Figures in the manuscript are too small and difficult to read in the PDF version. Please ensure that all images are replaced or resized for clarity.

Please submit a detailed response letter addressing each of the reviewers' concerns, along with a revised version of your manuscript that incorporates the necessary changes.

·

Basic reporting

These are the comments to be addressed
1. What is the need to resize the image to 768 by 768 pixels?
2. Why augmentation need? No justification in the manuscript.
3. The novelty of the study is missing

Experimental design

4.FastAI automates many tasks, which is great for beginners, but it can restrict more experienced users who want fine-grained control over certain operations. How to overcome the issue
5.FastAI adds a layer of abstraction over PyTorch, which sometimes introduces performance overhead due to extra layers of convenience features.

Validity of the findings

6. Apply TensorFlow with the same dataset, and compare it with the existing study.
7. These are the highly cited papers, that may be considered in the current study.

Additional comments

all the comments are suggested in above 3 section

·

Basic reporting

Language and Presentation:
The article is written in clear, professional, and unambiguous English, making it accessible to a wide range of readers, including those outside the immediate field of study. The terminology is used appropriately and well-defined according to the norms of academic writing. The structure follows PeerJ standards and discipline norms but is deviated thoughtfully to improve clarity and understanding.

Introduction:
The introduction is very effective in setting the stage, with an emphasis on the global importance of lung cancer and its dependency on histological examination for diagnosis. It presents a compelling case for the need for automated diagnostic tools, especially in pathology, where time and precision are of utmost importance. The motivation behind the study is explicitly stated, providing a solid foundation for the research.

Abstract:
The abstract well sums up the study's different parts: background, methodology, results, and implications in an organized and concise way. The rationale for using FastAI-2 with the ResNet-34 frame is clearly presented, focusing on the impressive accuracy of 99.78%. Hence, the abstract effectively and strongly conveys the objectives and the outcome of the study.
It is very informative and, thus, engaging for readers.

Literature Review and References:
The article clearly demonstrated a good grasp of the pre-existing literature and included references for relevant and current studies that have been done. By giving enough background on this proposed methodology, it presents the novelty and importance associated with it. The careful integration into the narrative of the background section strengthens the article's authenticity and situates the research within the global academic discourse.

Structure and Standards:
The structure of the article meets all standards of PeerJ. Logical flow is ensured through sections that are well differentiated, and reproducibility with transparency are crucial factors for high-quality research by outlining the description of the dataset and the details of methodology.

Conclusion and Future Directions:
The conclusion effectively summarizes the study's contributions, emphasizing the novel staining procedure and the impressive performance of the proposed model. The discussion of limitations, including dataset generalizability and the potential impact of staining variations, demonstrates a critical and reflective approach. The suggestions for future research provide clear pathways for extending the study's findings, showcasing the authors' commitment to advancing the field.

Experimental design

The manuscript's content is well aligned with the purpose and scope of the journal. It raises an important challenge in medical image analysis that can be resolved by using deep learning frameworks for automating lung cancer diagnosis.

Thorough Exploration
The research has been conducted to a high technical and ethical standard. The study incorporates a robust experimental framework, which includes data preprocessing, augmentation, model training, and evaluation. The dataset used is HIPAA-compliant, and the inclusion of references to principal investigators ensures credibility and transparency.

Methodological Clarity:
The article explains the methodology extensively and in great detail in a way that allows this study to be reproducible. The colour normalization is based on transformation in the Lab* colour space transformations; it provides enough detail that researchers wishing to replicate, modify, or extend would find it achievable. Information given on the preprocessing step and augmentation of data through FastAI-2 combined with ResNet34 architecture.

Data preprocessing and augmentation:
The discussion on data preprocessing is complete and emphasizes the need for it. The novel color normalization technique effectively reduces noise and low contrast in histopathological images and, therefore, is compatible with deep learning models. The augmentation strategy, using rotations and flips, contributes to diversity in the dataset and to the robustness of the model.

Evaluation and metrics:
The evaluation methods are well defined, using performance metrics like accuracy, precision, recall, F1-score, and ROC curves. The text clearly presents these metrics and supplements them with the use of visuals like confusion matrices and ROC curves, thereby enhancing understanding. The presentation of comparative results with other pre-trained models further confirms the effectiveness of the proposed framework.

Comparison and robustness:
The article compares the proposed ResNet34-based model with other state-of-the-art networks, thus providing a fair and transparent assessment of its strengths. The results show that the model not only has superior accuracy (99.78%) but also reduces execution time, making it a practical choice for clinical applications. The inclusion of misclassified image analysis (Figure 9) is commendable, as it provides insights into the limitations of the model and areas for manual intervention.

Citations and References: Sources are properly cited by giving due credit to the datasets, methodologies, and previous work. The way cited work is incorporated in the narrative is smooth; paraphrasing is used only when it is necessary to make the content not repetitive yet academic.

Validity of the findings

Validity of Findings
The argument is well-developed and logical within the article, and the paper does achieve the goals set at the introduction. The methodology for color normalization is clearly described, making sure that each step would be reproducible. Demonstrating a robust understanding of challenges in histopathological image processing, the approach is innovative to overcome challenges using Lab* color space for precise adjustments.

It effectively evaluates the results by integrating preprocessed images into a ResNet-34-based classification model, so that the experimentation provides experimental evidence for proving the correctness of the suggested technique.

Impact and Novelty
The novelty of this research is that it introduces a color normalization method called "color borrowing." Moreover, the study deals with significant noise and contrast in the LC25000 data set by using decorrelated Lab* color space; hence, it contributes substantially to the field of analysis of histopathology images.

Such integration of the proposed pipeline and FastAI-2 training framework for ResNet-34 is shown to be particularly impactful on deep learning applications. Although the manuscript does not evaluate the work's novelty in a quantitative manner to any competing methods, the thorough methodology behind it persuasively hints at its novation.

Replication and Contributions to the Field
The methodology is transparent and detailed, encouraging replication and further validation of the findings. The rationale for preprocessing histopathological images to make them compatible with machine-learning and deep-learning networks is well articulated, and the benefit to the field is substantial. By addressing image quality issues, the study facilitates better classification of lung histology, a critical area in medical diagnostics.

Conclusion
The conclusions are well-articulated and directly supported by the results. The manuscript indicates the effectiveness of the proposed approach toward lung histology classification while having an impact on their histology. Further in the conclusion, unresolved questions or limitations can be given explicitly, such as reliance on borrower images for colour normalization or potential difficulties for scaling this technique to larger sizes. Adding future directions like expanding the approach to other kinds of histological datasets gives it depth.

Experiments and Evaluations
The experiments are well thought out and adequately conducted. With ResNet-34 and FastAI-2, the paper establishes a high-performing classification framework. The choice of ResNet-34 is reasonable, as it achieves a good balance between complexity and efficiency. However, while the results are convincing, the manuscript could be improved by comparisons with more diverse models or ablation studies to evaluate the contributions of each preprocessing step.

Additional comments

The research article manifests a praiseworthy attempt to face the challenges of preprocessing histopathological images, more specifically the LC25000 dataset. A few additional positive comments based on the above-mentioned points are as follows:

Strength of Methodology
The authors have effectively addressed the issues with noise and contrast in biopsy images with the help of Lab* color space for decorrelation and normalization. The approach is new, novel, and really efficient towards image compatibility with deep learning models.
The process of describing step-wise the process of colour normalization facilitates clarity and reproducibility, an imperative condition to move further in the said direction of research.
This architecture, through the decision to choose ResNet-34 for integration by FastAI-2, presents a realistic combination whereby practicality combines both good classification and efficiency of computation.

Significance in the field
This solution, offered by this paper, is beneficial toward improving histopathological image quality, ensuring its effective and proper use for an accurate lung histology classification. Direct implications towards enhanced medical imaging diagnosis support tools come into mind.
The research opens up pathways for similar applications across other medical imaging datasets and domains by addressing the fundamental preprocessing challenges.

Presentation and Clarity
The manuscript is well-structured with a logical flow that guides readers from the problem statement to the experimental results. The inclusion of detailed equations, figures, and diagrams enhances understanding.

The authors have made proper explanations for technical concepts such as the conversion to Lab* color space along with its advantages so that the work is accessible to a broader audience, including researchers new to the field.

Scope of future research
Even though the method proposed is rigorously analyzed, it also serves as a springboard for further investigation. Including such a framework as FastAI-2 might indicate the potential of scalability and adaptability with a variety of datasets and classification problems.
The study represents a major direction of research: image preprocessing with deep learning that merges image preprocessing and medical diagnosis applications.

Recommendations for Improvement
To make the article more powerful, the authors can incorporate comparative analyses with other preprocessing methods, emphasizing the novelty and advantages of their approach quantitatively.

Adding discussions on unresolved questions such as reliance on borrower images or scalability of the method to high-resolution datasets would provide readers with a holistic view of the implications of the study.
In conclusion, discussing future directions may establish the study as a precursor to further progress in medical image analysis.

Final Remarks
The research article is a rich contribution to the field of medical image processing. It portrays a well-thought-out methodology and impressive results in image quality improvement and classification performance. In terms of technical rigor and innovation, the authors have demonstrated impressive work that will likely make a significant impact on both academic and clinical settings.

·

Basic reporting

Clearly mentioned the introduction and may include Literature Survey

Experimental design

No Comment

Validity of the findings

No Comment

Reviewer 4 ·

Basic reporting

The labels of your dataset have to be removed from the Introduction section. "lung_aca" is not a known or accepted abbreviation by practitioners for adenocarcinoma. This remark is valid for all labels.

Experimental design

Line 120: "We developed a deep learning model that effectively extracted meaningful information...". This phrase is in contradictory with the next phrase (line 122): "The utilization of FastAI-2 in conjunction with ResNet-34 helped to expedite the convergence process ...". Please clarify whether you have developed a deep learning model for your study or you have used a well-know classifier (ResNet-34).
Section "Proposed Methodology" line 165. The phrase: "This data is fed into the FastAI-2 framework for further classification ..." is somewhat vague and lacks scientific precision. I recommend specifying the exact process or method involved.
In the section titled "Classification Framework," a general theory regarding FastAI-2 and ResNet models is presented. However, this section is vague and lacks sufficient scientific detail. I recommend revising this section to include critical information, such as whether the deep learning model was pre-trained, the dataset used for pre-training (if applicable), and the details of the classification layer of the model.

Overall, the proposed method lacks sufficient detail and, in certain instances, is presented in a vague manner. Adding more specific information, by highlighting the novelty of your method and the updates/changes performed on the ResNet-34 mode, would significantly enhance the clarity and rigor of the manuscript.

Validity of the findings

The deep learning model used in the proposed method is clearly a multi-class classifier (with three classes as defined in the manuscript). However, at lines 308 - 311, binary performance metrics are presented. The use of binary metrics in a multi-class problem strongly suggests an evaluation or reporting flaw (As per Table 3, the accuracy is already at 0.96652 at the first epoch). I recommend revisiting the performance metrics and results, as the model appears to converge unusually quickly, requiring only 5 epochs for training.

Additional comments

All the images have to be redone or re-added. In the pdf version of the manuscript, the images are very small and very hard to read.

Reviewer 5 ·

Basic reporting

.

Experimental design

.

Validity of the findings

.

Additional comments

n general, the quality of the writing is not good, some sentences don't make sense and the writing quality needs to be improved.

Paragraph starting line 76, is duplicated.

Line 88: the sentence ending in "on this" seems incomplete, what does it rely on?

Line 104: the claim for FAST-AI makes approximation faster needs reference or data to back it up.

Line 118: is color normalization a new method or if not, what makes their method different from the past?



Line 146: The word conquered does not make sense here

line 148: what quality parameters?

Line 172: where does the data come from, a table with data configuration is needed, what data type, what disease, and so on

line 186: LC250000 is not introduced

line 200: if blue is dominant why do RG channels have high values?

Line 219: the Xs, Ys, and Zs are not explained



The color augmentation is not explained.



Line 253: Fast-AI is the first ML component that needs reference, what the author means by this sentence is not clear

Line 278: the exact preprocessing steps are not mentioned

line 340: did you set the epoch to 5? How can you make sure it is enough?



line 381: based on my understanding, the data is augmented and then it was used by the model, when you split the data to train and test, did split based on the case or based on image number, because the underlying data for both test and train may be the same case.


Overall there are major concerns with the paper that need to be addressed before it is ready for publication.

---

## Round 0.2 · accepted · Accept

Authors have clearly addressed all of the reviewers’ comments and the manuscript is ready for publication.

Reviewer 4 ·

Basic reporting

no comment

Experimental design

no comment

Validity of the findings

no comment

Additional comments

Thank you for taking into account my remarks.